# A reduced-complexity model for sediment transport and step-pool morphology

Matteo Saletti[1], Peter Molnar[1], Marwan A. Hassan[2], and Paolo Burlando[1]

[1]Institute of Environmental Engineering, ETH Zurich, Switzerland
[2]Department of Geography, The University of British Columbia, Vancouver, BC, Canada

*Correspondence to:* Matteo Saletti (saletti@ifu.baug.ethz.ch)

**Abstract.** A new particle-based reduced-complexity model ($CAST$) to simulate sediment transport and channel morphology in steep streams is presented. $CAST$ contains phenomenological parameterizations, deterministic or stochastic, of sediment supply, bed load transport, particle entrainment and deposition in a cellular-automaton space with uniform grain size. The model reproduces a realistic bed morphology and typical fluctuations in transport rates observed in steep channels. Particle hop distances, from entrainment to deposition, are well-fitted by exponential distributions, in agreement with field data. The effect of stochasticity both in the entrainment and in the input rate is shown. A stochastic parameterization of the entrainment is essential to create and maintain a realistic channel morphology, while the intermittent transport of grains in $CAST$ shreds the input signal and its stochastic variability. A jamming routine has been added to $CAST$ to simulate the grain-grain and grain-bed interactions that lead to particle jamming and step formation in a step-pool stream. The results show that jamming is effective in generating steps in unsteady conditions. Steps are created during high-flow periods and they survive during low flows only in sediment-starved conditions, in agreement with the jammed-state hypothesis of Church and Zimmermann (2007). Reduced-complexity models such as $CAST$ give new insights into the dynamics of complex phenomena such as sediment transport and bedform stability, and are a useful complement to fully physically-based models to test research hypotheses.

## 1 Introduction

The morphodynamics of steep gravel-bed rivers is characterized by complex feedbacks between sediment supply and storage (e.g Hassan et al., 2008; Hassan and Zimmermann, 2012; Recking, 2012; Recking et al., 2012), bed load transport and flow resistance (e.g. Yager et al., 2007; Recking et al., 2008) and a rather stable bed morphology with a variety of bed surface structures (see reviews by Comiti and Mao, 2012; Rickenmann, 2012; Church and Ferguson, 2015). The traditional sediment transport capacity approach (Wainwright et al., 2015), which has been developed for low-land streams, performs poorly in steep fluvial systems. Among other reasons, in steep channels the threshold of motion varies with slope, local bed structures and antecedent flood events (e.g. Lamb et al., 2008; Turowski et al., 2011; Scheingross et al., 2013; Prancevic and Lamb, 2015), a power-law relation between fluid shear stress and sediment transport yields orders of magnitude differences between measurements and predictions (e.g. Rickenmann, 2001), and the presence of macro-roughness elements (such as boulders and

log-jams), whose size is comparable with the water depth, makes calculations of flow resistance extremely complex (Yager et al., 2007; Schneider et al., 2015).

The step-pool morphology is commonly encountered in mountain catchments at slopes grater than 3% (Montgomery and Buffington, 1997; Comiti and Mao, 2012), where large boulders and woody debris create channel-spanning structures called steps, with pools immediately downstream formed by the scouring effect of the tumbling water flow (see reviews by Chin and Wohl, 2005; Church and Zimmermann, 2007). Step-pool channels have been studied extensively in order to understand under which conditions they are formed, remain stable and are eventually destabilized (e.g. Abrahams et al., 1995; Curran, 2007; Zimmermann et al., 2010). This topic is still an open issue and, despite observations of a certain degree of regularity in step-pool geometry (e.g. Chartrand et al., 2011), it has been recognized both in the field (e.g Zimmermann and Church, 2001; Molnar et al., 2010) and with lab experiments (e.g. Curran and Wilcock, 2005; Zimmermann et al., 2010) that there is no single mechanism behind step formation and collapse (Curran, 2007). In fact, we are of the opinion that step formation and stability should be treated as stochastic processes, which result in largely random locations of step-forming boulders usually referred to as keystones (Church and Zimmermann, 2007; Zimmermann et al., 2010). A hypothesis on step stability has been proposed by Church and Zimmermann (2007) and tested experimentally by Zimmermann et al. (2010). The authors suggested a similarity between step formation and granular phenomena by postulating that steps are inherently more stable than predicted by the Shields diagram because they are arranged in a jammed state, which occurs in granular flows (for a review on the jamming phenomenon see Liu and Nagel, 2010). They proposed a diagram where the likelihood of finding stable steps in a channel is dependent on 3 parameters: (1) the jamming ratio (the ratio between the channel width $W$ and the $d_{84}$ of the surface), (2) the transport stage (the ratio between the applied shear stress $\tau$ and the critical shear stress $\tau_{cr}$), and (3) the sediment concentration (the ratio between sediment supply $Q_S$ and water discharge $Q$). So far this theory has been tested against lab experiments and field data (Zimmermann et al., 2010) but to our knowledge step formation and collapse have not yet been explicitly modelled. Understanding the conditions under which step-pool sequences are stable is of major practical importance because step collapses and consequent boulder mobilization can severely impact human infrastructures causing natural hazards (e.g. Badoux et al., 2014). Moreover, artificial step structures are often built in alpine rivers as energy dissipators and for erosion control; therefore, their stability needs to be carefully assessed.

Physically-based modelling of flow and sediment transport in steep mountain streams in mobile bed conditions is impractical because (a) the flow field over the rough bed is very complex; (b) single-grain mobility is impossible to solve; (c) long-term simulations are required to develop a dynamically changing channel bed. An alternative to fully physically-based modelling is that of reduced-complexity models. Instead of solving differential equations of flow and sediment transport, RC models formulate physically-meaningful local flow-grain interaction rules with very few parameters in a cellular automaton space. The reduced-complexity (RC) framework has also been applied successfully in fluvial geomorphology (Nicholas, 2005) as a learning tool to gain new insight into the temporal and spatial dynamics of complex systems in general (Goldenfeld and Kadanoff, 1999; Paola and Leeder, 2011; Rozier and Narteau, 2014; Tucker et al., 2015). Since the classical cellular model of Murray and Paola (1994) which effectively captured the main patterns of river braiding, RC models have been used to describe geomorphic phenomena, such as riverbank failure (Fonstad and Marcus, 2003), bedrock cover (Hodge and Hoey, 2012), river

avulsion (Jerolmack and Paola, 2007), sand dunes (Narteau et al., 2009), river deltas (Seybold et al., 2009; Liang et al., 2015), patterns of erosion-sedimentation (Crave and Davy, 2001; Chiari and Scheidl, 2015), transport in gravel bed-rivers (MacVicar et al., 2006) and landscape evolution (Coulthard and Wiel, 2007; Van De Wiel and Coulthard, 2010).

In this paper we present a new reduced-complexity stochastic model for step-pool streams based on grain-grain interactions: $CAST$ (Cellular Automaton Sediment Transport). $CAST$ simulates a generic fluvial channel on a cellular-automaton domain, where the bed is formed by an arrangement of particles like in a sandpile model (e.g. Bak et al., 1988; Kadanoff et al., 1989). The basic processes of bed load transport, particle entrainment and deposition are treated at the grain scale taking advantage of analogies between bed load transport and granular phenomena (e.g. Frey and Church, 2009, 2011; Houssais et al., 2015). The stochastic framework of $CAST$ has two main reasons. First, the goal of the model is not to predict deterministically the morphology of a specific river reach but rather to capture feedbacks related to its dynamics and to test research hypotheses on step formation and stability. Second, both bed stability (Zimmermann et al., 2010) and bed load transport (Einstein, 1937, 1950) have been recognized to be stochastic processes, and recent approaches to sediment transport have successfully followed this framework (e.g. Turowski, 2010; Furbish et al., 2012; Heyman et al., 2013; Ancey and Heymann, 2014; Armanini et al., 2015).

The paper objectives are: (a) to present a new reduced-complexity model that simulates bed load transport and channel morphology at the grain scale and to test the effect of different parameters and stochastic forcing on the model outcomes; (b) to explore the effect of jamming on sediment transport and step formation, in comparison with the framework of the the jammed-state hypothesis of Church and Zimmermann (2007). The paper is organized as follows. First, the model rationale and parameters are presented. Second, the effect of different parameter sets on the model outcomes are explored in steady-state simulations, and the effect of the stochasticity on different variables is shown. Then, the process of jamming is parameterized and its role on step formation and stability is explored in unsteady simulations. Finally the results are discussed and compared to the jammed-state hypothesis.

## 2    Model Rationale

$CAST$ operates in 2-D cellular-automaton space, which is a rectangular grid of constant length $(X)$ and width $(Y)$ corresponding to a generic river reach (see Fig. 1). The domain is discretized such that all the dimensional quantities are expressed as multipliers of particle size $d$. The model developed in this paper works with uniform-size particles, so $d$ is also equal to the dimension of a cell. For example, a simulation domain having $X = 300d$ and $Y = 20d$ represents a river reach with an average length equal to 300 median diameters and an average width equal to 20 median diameters.

Particles in the model domain can be either on the bed or in motion, i.e. part of the bed matrix or of the transport matrix (Fig. 1). The bed matrix $Z$ is composed of particles piled one above the other like in a sandpile model (e.g. Bak et al., 1988). The local bed elevation at a generic location $(i, j)$, where $i[1 : X]$ is the index for the longitudinal coordinate and $j[1 : Y]$ is that of the transversal coordinate, is given as the total number of particles $Z_{i,j}$. Particles can leave the bed matrix as a result of

entrainment and can enter the bed matrix as a result of deposition. In the case of entrainment and deposition the local value of elevation $Z_{i,j}$ is reduced or increased by one grain unit $d$, respectively:

$$Z_{i,j}(t+1) = Z_{i,j}(t) \pm d \tag{1}$$

Particles in motion are allocated to the transport matrix $TR$, which consists of two layers, and they move as bed load, i.e. they are in contact with the bed and interact with it. They also interact with each other by collisions. Particles move with a constant velocity, one cross-section downstream for every time step. In this way particle velocity $v_p$, particle size $d$ and time step $\Delta t$ are connected:

$$v_p = \frac{d}{\Delta t} \tag{2}$$

Although in simulation $\Delta t$ is a unitless time, Eq. 2 together with the grain and domain size give it a physical meaning connected to particle velocity.

Particles enter the system with a specified input rate $I_R$ which is the number of particles entering the system at the upstream boundary for every time step, and they leave the system as output rate $O_R$ at the downstream boundary. In analyzing the spatial output of the model we consider only a reduced part of the domain which we will call hereafter the control volume, excluding the first 10 cross-sections upstream and the last 10 downstream, to avoid the influence of the upstream and downstream boundary conditions (see next sections).

## 2.1 Model Components

### 2.1.1 Sediment Input

The first parameter of $CAST$ is the particle input rate $I_R$ or the specific input rate $i_R$, defined as the total input rate $I_R$ divided by the channel width $Y$; $i_R$ can assume values in the interval $[0:1]$ because no more than one particle can enter a single cell at the upstream boundary in $\Delta t$. The supply of particles to the system can be treated as constant input by specifying a value of $i_R$ for the entire simulation or as variable random input by specifying a mean value $\overline{i_R}$ with a temporally-variable term $i'_R(t)$, uniformly distributed around $\overline{i_R}$. In $CAST$, $i_R$ represents a generic amount of sediment which is delivered to the channel from all the possible sources (alluvial transport, colluvium activity, bank erosion, etc) rather than a specific transport rate in a given cross-section. The actual input to the system can be considered to be the transport rate measured in the first cross-section of the control volume.

### 2.1.2 Sediment Transport

Particles are transported as bed load along the channel with a constant velocity (see Equation 2). A particle can join the transport matrix $TR$ when it enters the system as input or once it has been entrained from the bed. A particle can also leave the transport

matrix $TR$ when it moves beyond the last cross-section, becoming part of the sediment output $O_R$, or when it deposits on the bed surface. The maximum number of particles being transported from one cross-section to the next one is equal to $2Y$, i.e. 2 times the channel width, because the $TR$ matrix has two layers.

Particles move preferentially directly to the downstream cell (90% probability), with a small chance for lateral displacements (5% to the left and 5% to the right). This aims to represent a grain having a transport vector aligned with the dominant flow direction with limited diversion. In absence of observations, we constrain the probability for lateral movements in a reasonable interval of 10%. The model is not sensitive to lateral dispersion, at least in the parameter space we have tested. Along its path, a particle can collide with another particle in transport or collide with one of the two boundaries (left and right banks). In both cases the collision leads to the loss of momentum and cessation of motion and the particle deposits on the bed (see section 2.1.4). When a particle deposits on the bed, it changes the local roughness but without directly displacing other particles, i.e. the $CAST$ does not account for collective entrainment as described by Ancey and Heymann (2014).

Sediment flux in the model, $q_S$, is computed at the total number of particles in $TR$ in the control volume divided by the domain size, i.e. $Y \cdot (X - 20)$. This specific rate $q_S(t)$ is computed for every time step.

### 2.1.3 Particle Entrainment

The key process in $CAST$ is the particle entrainment which is considered to be dependent on the local bed topography and on the flow conditions. The degree of exposure of particles on the bed has been shown to strongly influence sediment entrainment and transport especially in steep streams (e.g Kirchner et al., 1990; Malmaeus and Hassan, 2002; Yager et al., 2012; Prancevic and Lamb, 2015). Moreover many feedbacks exist between bed roughness, flow resistance and particle mobility and transport (e.g. Recking et al., 2008; Wilcox et al., 2011) which makes it reasonable to consider particle entrainment as a stochastic process.

The effect of local topography on entrainment is accounted for by calculating the local relative exposure $R$ of a particle on the bed. For a generic cell $(i, j)$ in the domain, the relative exposure $R_{i,j}$ is given by the difference between the elevation of the cell $Z_{i,j}$ and the average elevation of the neighboring cells along the flow direction (the 2 in the same cross-section and the 3 downstream):

$$R_{i,j} = Z_{i,j} - \frac{Z_{i,j-1} + Z_{i,j+1} + Z_{i+1,j-1} + Z_{i+1,j} + Z_{i+1,j+1}}{5} \tag{3}$$

In the case of a cell located close to one of the banks (i.e. when $j = 1$ or $j = Y$), $R_{i,j}$ is evaluated considering the cell in the same cross-section and the 2 cells downstream.

Entrainment is based on $R$ exceeding a threshold $R^*$ for entrainment. The probability of entrainment $p_E$ is then defined as $p_E = P_r[R \geq R^*]$. $CAST$ can model the entrainment process as deterministic or stochastic (see Fig. 2). In the deterministic case, the threshold is a constant $R^* = E$ and the probability of entrainment is:

$$p_E = P_r[R \geq E] = \begin{cases} 0 & R < E \\ 1 & R \geq E \end{cases} \tag{4}$$

5      In the stochastic case, the threshold $R^*$ is modelled as a random variable with a logistic probability density function $f(R^*)$ and a cumulative distribution function $F(R^*)$:

$$f(R^*) = \frac{e^{-\frac{R^*-E}{S}}}{S\left[1+e^{-\frac{R^*-E}{S}}\right]^2}$$
$$F(R^*) = \frac{1}{1+e^{-\frac{R^*-E}{S}}} \tag{5}$$

The distribution has a mean $\mu_{R^*} = E$ and a variance $\sigma^2_{R^*} = \frac{\pi^2 S^2}{3}$. In this way the entrainment probability $p_E$ of a cell with relative exposure $R$ depends on two parameters: the mean value of the threshold distribution $E$ and the variability in the 10   threshold $R^*$ proportional to $S$:

$$p_E = P_r[R \geq R^*] = F(R) = \frac{1}{1+e^{-\frac{R-E}{S}}} \tag{6}$$

Figure 2 shows the deterministic and stochastic parameterizations of entrainment.

     Conceptually, the value of $E$ is inversely related to the magnitude of the flow. Large $E$ means low probability of entrainment, typical of low flow conditions and low shear stress, while small $E$ means high $p_E$ for the same relative exposure values and so 15   high flow conditions and high shear stress.

### 2.1.4   Particle Deposition

Particles in transport (i.e. belonging to the $TR$ matrix) can be deposited in three cases: (a) when they collide with another particle in motion; in this cases both particles involved in the collision events are deposited, each one in its present location; (b) when they collide with one of the two channel banks; (c) because of their interaction with the bed surface.

20      The relation between particle deposition and bed surface is modeled using the relative exposure matrix $R$: particles in motion deposit in areas of local depressions, i.e. cells with $R < 0$. The deposition process is treated as deterministic, with a threshold function having a fixed value of $-0.5$ below which the probability of deposition $p_{dep} = 1$. This simplification allows on one hand to avoid redundant parameters poorly connected to physical processes, and on the other hand to shift the variability in sediment transport to the entrainment process which has a more straightforward relation with hydraulics and local channel bed 25   topography. Case (a) and (b) are much less common than (c) since they both require lateral movement and the presence of an obstacle (i.e. another particle or the channel banks).

### 2.1.5 Boundary and Initial Conditions

$CAST$ needs one boundary condition for the lateral banks and one for the downstream boundary at the channel outlet. The boundary condition for the banks is deposition when a moving particle in the $TR$ layer collides with one of the two lateral boundaries. The boundary condition for the last cross-section at the downstream boundary is given by fixing its elevation $Z(X,j) = 0$. This is equivalent to a control section with a fixed elevation (e.g. a check dam or a weir) somewhere downstream. To minimize the effect of this boundary condition on the model outcomes, all spatial variables are computed only over the control volume, which is a reduced portion of the entire channel (see Fig. 1).

In order to avoid long simulation times required to fill the channel with particles, we start every simulation from an initial slope, slightly less than the equilibrium slope, with random noise. The model in not sensitive to this initial condition, i.e. the final equilibrium slope is only a function of the chosen set of parameters (mainly the input rate $i_R$ and the entrainment parameter $E$). Different initial conditions determine how long the system needs to reach this equilibrium state.

### 2.2 Rough-Bed and Jamming in $CAST$

$CAST$ operates in two modes, with and without dynamic jamming. The rough-bed model without jamming ($CAST_{RBM}$) simulates a generic rough-bed channel where processes of transport, entrainment and deposition are considered regardless of any additional granular effect (except for particle collisions and deposition after collisions with the channel banks). The jamming model ($CAST_{JM}$) simulates explicitly the process of jamming (blocking) when the density of particles transported in the same cross-section exceeds a threshold. In this case particle interactions lead to deposition of all grains in that cross-section on the bed. This blocking process is considered permanent, i.e. the jammed particles are locked into channel width spanning structures which cannot be entrained anymore. In the same cross-section, entrainment and deposition of other particles is still possible, except for those grains that have been subjected to jamming. This parameterization aims to represent in a simplified way the additional force chains that keep grains together around keystones, as shown in the jammed-state hypothesis of Church and Zimmermann (2007). Intuitively and similarly to other phenomena where jamming is common (e.g. in hoppers) we set up the jamming threshold equal to the channel width $Y$. In a one grain-size model like $CAST$ this implies that jamming is happening when the transport layer is full of particles (one entire cross-section full of transported particles).

For every time step, the computation sequence is as follows: (1) Sediment input enters the system in the first cross-section. (2) Particles in transport move one cell downstream (straight, left or right), if they collide with other particles or with one of the banks they deposit, otherwise they remain in transport. In the case of $CAST_{JM}$ the jamming condition is checked for every cross-section. When the number of particles traveling in the same cross-section exceeds the jamming threshold all grains are deposited on the bed and frozen. (3) For every particle in motion the condition for deposition is checked: if it is satisfied the particle leaves the transport matrix and is deposited on the bed. (4) For every particle in the bed matrix (except for those jammed in stage 2 and deposited in stage 3) the condition for entrainment is checked: if it is satisfied the particle leaves the bed matrix and joins the transport matrix. (5) The boundary condition at the channel outlet is applied.

## 3 Model Set-Up: Steady-State Simulations

To set-up the model, we have run a set of simulations using $CAST_{RBM}$ with (a) stochastic entrainment with constant $E$ and $S$ parameters, and (b) constant specific input rate $i_R$. The domain is $Y = 20d$ and $X = 300d$. The domain size has been chosen not to represent or scale any specific river channel, but rather to observe the features and test the hypotheses we are interested in, at a reasonable computational cost. Simulations with a much larger scale (up to 2 orders of magnitude) have also been performed and we observed no significant scale effect on the final outcome (see Supplementary material). The simulations were run until a steady state was reached. This condition is achieved when for a given combination of $i_R$ and $E$ the channel slope does not change, i.e. the point at which the particle count (stored sediment volume) in the channel reaches a steady state and the sediment output is on average equal to the input. We explored the effect of different input rates $i_R$ and the entrainment parameters $E$ on the final bed structure. Moreover, since $CAST$ is a stochastic model, we perform 20 realizations for every set of parameters to quantify stochastic variability. The set of parameters used in these steady-state simulations is shown in Tab. 1.

We analyzed the results in terms of:

- storage volume $V$: the total number of particles in the bed matrix. The time series of this variable indicates if the channel is in a phase of aggradation ($V$ increasing in time), degradation ($V$ decreasing in time) or equilibrium ($V$ constant in time on the average). It also indicates massive sediment evacuation events;

- specific sediment flux $q_S$: the number of particles in motion per unit length and width in the control volume;

- mean relative exposure $< R >$: the spatially-averaged value of $R$ of all the cells in the control volume domain evaluated with Eq. 3; and standard deviation of the relative exposure $\sigma_R$;

- particle hop distance $HD$: the step length of a single particle from the point it is entrained (or enters the channel) to the point it is deposited or exits the channel.

### 3.1 Storage Volume, Sediment Transport, Bed Morphology, and Hop Distances

The main outputs of $CAST_{RBM}$ in a steady-state simulation ($E = 1.5$; $i_R = 0.5$) at equilibrium are shown in Fig. 3. First, the storage volume (Fig. 3a) exhibits a dynamical equilibrium: the volume oscillates, alternating phases of aggradation and degradation, because the stochastic parameterization of particle entrainment leads to a continuous exchange between the bed and the transport layer. Second, the times series of sediment transport (Fig. 3b) show that even with a constant input rate, sediment transport fluctuates as observed in the field and in the lab (e.g. Recking et al., 2009; Saletti et al., 2015). Moreover, $CAST_{RBM}$ produces a realistic rough-bed morphology (Fig. 3c), with the mean $R$ and standard deviation $\sigma_R$ being a function of the input rate $i_R$ and the entrainment parameter $E$. The input rate and the entrainment parameter also determine the final slope of the channel.

One of the advantages of a RC models like $CAST$ is that it is possible to track the movement of every single particle in the system and so to compute all particle step lengths (measured from entrainment to deposition). This is an important quantity

which, since Einstein's probabilistic theory on bed load transport (Einstein, 1937, 1950), needs to be reproduced by any reliable particle-based transport model. Since the word 'step' in this paper refers to the bed structures created by grain arrangements, for the sake of clarity we will call hereafter, following Furbish et al. (2012), particle hop distances ($HD$) what in literature is usually called step length, i.e. the distance travelled by a particle from entrainment to deposition (Fig. 3d). For every simulation we computed values of $HD$ for all the particles and we find they follow an exponential distribution. In Fig. 4 we show for 4 different combinations of $i_R$ and $E$ the probability density functions of $HD$ and the exponential fit. The good fit given by this distribution is in agreement with previous studies dealing with particle travel distances (e.g. Hill et al., 2010; Hassan et al., 2013; Schneider et al., 2014). Since in the model no $HD$ distribution is specified a priori, the agreement shows that the phenomenological rules of particle entrainment, transport, and deposition of $CAST$ are realistic. The values of $HD$ obtained in our simulations are always much smaller than the channel length $X$ (e.g. the maximum observed $HD$ is less than $X/2$), therefore the system scale is not influencing our results. The relation between $HD$ and the model parameters is explored in the next section.

## 3.2 Role of Input Rate and Entrainment Probability

With the steady-state simulations we explored the effect of changing input rate and entrainment parameter on the model outcomes. These two parameters are important because they can be linked to the jammed-state diagram parameters of Church and Zimmermann (2007). The input rate $I_R$ is related to the sediment concentration $\frac{Q_S}{Q}$, which quantifies the effect of sediment supply on step stability. The entrainment parameter $E$ determines the entrainment probability and is directly related to the transport stage $\frac{\tau}{\tau_{cr}}$ which quantifies the effect of the hydraulic forces on step stability.

Some of the simulations, characterized by low input rate and high entrainment probability ($E = 1$, $i_R < 0.4$), yield what we call 'washed-out' case, i.e. the bed matrix remains empty. This represent a limiting case where hydraulic forces are too high and sediment supply is too low to be able to sustain a fluvial channel. This constrained our parameter space to 27 simulations in which a channel was formed and maintained around an equilibrium point.

The stochastic simulation of 20 realizations of each of the 27 parameter sets showed that the mean storage volume $\overline{V}$ and the mean relative exposure $< R >$ converged to the same values. Also the mean hop distances $< HD >$ and the standard deviations of the relative exposure $\sigma_R$ did not change significantly.

The values of 4 key variables for the 27 simulations (parameter combinations), averaged over the 20 realizations, are shown in Fig. 5. The mean relative exposure ($< R >$ in Fig. 5a) is obtained by a spatial average of all the values of $R$ for a given time step, and then temporally averaged over the last 20000 time steps in the equilibrium phase. $< R >$ is directly related to the slope of the channel and the storage volume. It increases with increasing input rate and increasing entrainment parameter: channels with large sediment supply and low entrainment probability are those with larger $< R >$ and larger storage volumes. The same trend can be inferred by looking at the mean standard deviation of $R$ ($\sigma_R$ Fig. 5b), also obtained as a spatial average over the equilibrium phase for every time step.

Specific sediment flux $q_S(t)$ is on average equal to the input rate $i_R$ at equilibrium, but fluctuating around its mean value, as shown in Fig. 3b. The degree of memory of these fluctuations is captured by the Hurst exponent $H_{q_s}$, whose mean value for

the steady state simulations is shown in Fig. 5c. The values of $H_{q_S}$ obtained in all the realizations of all the simulations are consistent with those obtained from flume experiments by Saletti et al. (2015), being in the interval $[0.5:1]$. This identifies a long-memory regime which is stronger in the model ($H \to 1$) when the entrainment probability is high (low $E$) and the input rate is low. $H_{q_S}$ shows large variability in different realizations, although its mean value shows a clear trend with both $i_R$ and

$E$ (Fig. 5c).

The mean particle hop distances ($< HD >$ in Fig. 5d) display two contrasting trends. For values of $E \leq 1.5$ (high entrainment probability) $< HD >$ is decreasing consistently for increasing input rates $i_R$ as a consequence of large particle activity (collisions between particles become very frequent). For larger values of $E$ (low entrainment probability) the maximum of $< HD >$ is for average values of $i_R$ (around 0.4). For $i_R < 0.4$ there is a stronger interaction with the bed (which has larger

$< R >$ and $\sigma_R$ for larger E, as shown in Fig. 5a,b) and so more likelihood of particle deposition, whereas, for large $i_R$, collisions again dominate hop distances because of large particle activity. In both cases this leads to a reduction of $< HD >$.

### 3.3  $CAST_{JM}$: Particle Jamming

The analysis above showed that $CAST_{RBM}$ is able to reproduce fluvial channels with a spatially variable rough bed and to capture basic and important physical phenomena, such as the variability in sediment flux and the exponential distribution of

particle hop distances. However, to simulate the formation and test the stability of steps, we need to take into account the effect of particle jamming. This is a well-studied phenomenon in granular physics that has been advocated to be essential in the step stability process and considered through the jamming ratio (the ratio between the channel width and $d_{84}$ of the surface) in the diagram proposed by Church and Zimmermann (2007). In our reduced-complexity model $CAST_{JM}$ we account for the jamming effect by blocking particles when local sediment concentration exceeds the jamming threshold, and depositing them

in permanent structures on the bed.

Jamming simulations were run with the same parameter sets of the steady-state rough-bed model case. Three different situations occur:

1. When particle activity is too low (low sediment transport) the jamming threshold is rarely (often never) exceeded.

2. When particle activity is too high (high sediment transport) jamming is occurring too often in time and space, and the
storage volume of the system keeps increasing because of the large amount of particles depositing upstream of the step structures. As a result an equilibrium channel is never reached.

3. When particle activity is in-between the two previous situations, jamming is occurring at a rate which allows the formation of steps and maintains an approximately equilibrium channel.

The first situation represents the rough-bed case discussed previously. The second one represents a case which is very

unlikely to happen in river systems where fluvial sediment transport is rarely going to exceed the jamming threshold and certainly not for very long periods of time (e.g. only during large flood events). For the purpose of this study we focus on the last situation where jamming is effectively creating steps. When particles are jammed and instantly deposited, they trigger

a deposition process which is propagating upstream, since the values of relative exposure $R$ will be immediately reduced. This represents what happens in natural step-pool systems, where steps are created, among other factors, by deposition and clustering of sediment around large boulders called keystones, and deposition between steps continuously changes the channel (e.g. Molnar et al., 2010).

We show the effect of adding jamming to the model by comparing simulations having the same parameter sets and same initial conditions ($i_R = 0.5$ and $E = 1.25$) in $CAST_{RBM}$ and $CAST_{JM}$ runs. The cumulative number of jammed cross-sections shows that jamming is a rather intermittent phenomenon with many long periods of no jamming (Fig. 6a). At the end of this simulation 29 cross-sections were jammed (around $10\%$ of the total). The longitudinal profiles of bed elevation (Fig. 6b) show how $CAST_{JM}$ is able to create step structures and this increases the total slope of the channel and its storage, even

if the slope between steps is the same as in the case without jamming ($CAST_{RBM}$). The boxplots of the instantaneous (i.e. calculated for every time step) values of $R$, both the mean $< R >$ (Fig. 6c) and the standard deviation $\sigma_R$ (Fig. 6d), show that the model with jamming yields a rougher and more variable bed. Instantaneous values of specific sediment flux $q_s$ (Fig. 6e) show that jamming slightly increases the variability in $q_S$ and prevents the formation of an equilibrium slope (which would imply $q_S \simeq i_R = 0.5$) because the system is still aggrading and increasing its storage. Finally the values of the Hurst exponent

of sediment flux $H_{q_S}$ (Fig. 6f) for the 20 realizations clearly plot separately in the case with and without jamming, with the latter having much greater values. This longer-term memory is likely due to a combination of sediment pulses created by step collapses and the weak but present trend towards aggradation in the $CAST_{JM}$ simulations.

## 4    The Effect of Stochasticity

The processes of particle entrainment and sediment supply can be parameterized as deterministic or stochastic. In the simula-
tions presented in the previous sections we used a stochastic parameterization for particle entrainment and a constant sediment supply. To explore the effect of stochasticity on the model results, in the next two sections we quantify the effect of stochasticity in entrainment and sediment supply explicitly.

### 4.1    Stochasticity in the Entrainment

The entrainment probability in $CAST$ can be parameterized as a deterministic or stochastic process (Section 2.1.3). A stochas-
tic parameterization allows a degree of variability in the entrainment threshold and can be controlled by two parameters ($E$ and $S$), while the deterministic parameterization has a unique entrainment threshold $E$ (Fig 2).

The comparison for a simulation with $i_R = 0.5$, $E = 1$ and $S = E/5$ is shown in Figure 7. When the entrainment process is treated as stochastic, the variability of sediment flux is much larger both in case with and without jamming (Fig. 7a). This is due to the fact that when the channel has reached equilibrium in the deterministic case the interaction between the bed and the
transport is very low. All particles below the threshold stay on the bed and those above the threshold are entrained. The reduced particle activity can be inferred also by looking at the distribution of particle hop distances (Fig. 7b). In the deterministic case the distributions are shifted towards larger values because particles interact much less with the bed and travel further

downstream. The effect of modeling the entrainment as a deterministic process on the bed morphology itself is that the final configuration of the channel in the threshold case is much steeper (cumulative distribution functions of $R$ plot towards larger values in the T-case) since no entrainment is possible below the threshold: the channel can bear steeper slopes and store more sediment (Fig. 7c). However, this does not translate into a rougher surface; $\sigma_R$ shows that channels where the entrainment is modeled with a threshold function have very low variability around $< R >$: they tend to look more like steep and uniform ramps than like realistic fluvial channels (Fig. 7d).

This analysis support our choice of modeling the entrainment as a stochastic process. This is not only more physically reasonable because the process of particle displacement is random per se, but it is not possible to obtain a realistic rough-bed morphology in a reduced-complexity model like $CAST$ without a stochastic parameterization of particle entrainment.

## 4.2   Stochasticity in the Input Rate

The effect of stochasticity in the input rate $i_R$ is shown on simulations with $E = 1.5$ and constant $i_R = 0.5$, or random input rate uniformly distributed around the mean value $< i_R >= 0.5$.

The effect of stochasticity in the input rate is much smaller than that of entrainment. The distributions of sediment flux (Fig. 8a) almost overlap both in the case with and without jamming. Like in the entrainment case, jamming causes a constant increase in sediment storage (and so the median transport rate is slightly below the equilibrium value of 0.5). An overlap of the 4 simulations is also observed when looking at the distributions of particle hop distances and final $R$ (Fig. 8b and c). The simulations with and without jamming plot separately only in the case of $\sigma_R$ (Fig. 8d). The larger values of $\sigma_R$ in the case of variable $i_R$ and jamming are due to the fact the variability in the input facilitates the jamming process and increases the relative exposure.

These results highlight that the variability and the fluctuations observed in the sediment output variables of the model do not depend on the variability of the sediment input, but are instead function of the internal dynamics of the system, given by the local grain-grain and grain-bed interactions. In other words $CAST$ acts as a shredding filter of the input forcing (Jerolmack and Paola, 2010; Van De Wiel and Coulthard, 2010).

## 5   Unsteady Simulations

Although jamming is effective in generating a step-like morphology under certain steady-state sediment input and entrainment conditions, we recognize that step formation is an intermittent process in which flow variability in time is important. Typically step-pool sequences are partially or totally destroyed during large flood events and then reworked and stabilized during the following low flows periods (e.g. Lenzi, 2001; Turowski et al., 2009; Molnar et al., 2010). We show the effects of changing flow conditions by simulating 4 consecutive floods of equal magnitude (Fig. 9a). In $CAST$ the hydraulic conditions are represented by the entrainment parameter $E$. Therefore, to simulate a change in the flow, we modify the value of $E$ to represent two extreme cases (Fig. 9b): low flow with $E = 2$ (low entrainment probability) and high flow with $E = 1$ (high entrainment probability). Moreover, we explore two different situations: (1) we keep the input rate $i_R$ constant, incorporating all the effects

of the unsteadiness in the entrainment parameter $E$ (Case I in Fig. 9c), and (2) we change also the input rate $i_R$ in response to changes in the flow conditions (Case II in Fig. 9d). To facilitate the comparison, the total sediment input over the entire simulation is the same in Case I and II. To mimic the rising and falling limb of an hydrograph, both $E$ and $i_R$ were increased and decreased gradually. The relevant parameters of these unsteady simulations are summarized in Tab. 2. Runs were performed
both with and without jamming (i.e. with $CAST_{RBM}$ and $CAST_{JM}$), to check if and when steps are formed and how many of them remain stable.

## 5.1   Sediment Storage and Sediment Transport

The temporal pattern of storage volume and sediment flux in the unsteady simulations is shown in Fig. 10. The volume for the rough-bed case (in blue) clearly displays phases of degradation during high flow and aggradation during low flow. Without
jamming the channel tends to erode during high flows when the entrainment probability is high and to gain sediment again during low flow when the entrainment probability decreases. This turnover is more evident when the input rate is constant (Fig. 10a). With the effect of jamming the picture changes. During high flows the mobile grains are trapped in the channel in steps, while during low flows the channel increases its storage because of grain deposition between steps and channel infilling. With a variable input (Case II), jamming creates more steps and increases the storage volume which then remains constant during
the following low flow phases because of the reduced input rate and low entrainment probability (Fig. 10b).

The specific sediment flux when the input is constant (Fig. 10c) shows a large variability for the rough-bed case responding to changes in $E$ during low and high flow conditions. In the jamming case, the response to the change in flow conditions is also present but the jamming process modulates the sediment flux towards the equilibrium conditions rapidly. When the input varies with flow conditions (Fig. 10d), the rough-bed model yields the same pattern as the case with constant input with the difference
that here the equilibrium condition is changing during low and high flow (0.3 and 0.6 respectively). In the jamming model instead, the sediment flux is almost instantly in equilibrium with the input rate during low flows, while during high flows, the large input rate, together with the high entrainment probability, causes so many jamming events that inhibit the system to reach an equilibrium state and the channel keeps increasing its storage. We show the statistical distributions of specific sediment flux for the 4 cases in Fig. 11. It can be seen that the distributions are centered around the equilibrium point of 0.4 (especially the
jamming case in red for Case I), with the rough-bed model having a more spread function due to the more intense phases of aggradation and degradation. The distributions of Case II (dashed lines) are clearly bimodal because of the two equilibrium sediment input rates (0.3 and 0.6).

## 5.2   Step Formation and Stability

The unsteady flow also has impacts on bed roughness in $CAST$. The time series of the standard deviation $\sigma_R$, which represent
the degree of roughness of the bed, is shown in Fig. 12 for the unsteady simulation with constant and variable input rate. In both cases jamming produces a rougher surface during high flow which is a indication that step structures, causing a larger departure from the mean $R$, are being formed. When the input is constant (Fig. 12a), $\sigma_R$ goes back to value of low flow for all the 4 floods, because steps that were formed are being buried by sediment. When instead the input rate is reduced during low

flow to simulate sediment-starved conditions (Fig. 12b), $\sigma_R$ decreases but not to its pre-flood value because many of the steps created during high flow can survive and do not get buried in-between floods.

The same can be inferred from the longitudinal profiles of bed elevation of the simulations with jamming (Fig. 13). At the end of every high-flow period, the longitudinal profile shows a stepped morphology due to jamming. In the following low-flow periods the steps were buried in Case I (having input rate $i_R = 0.4$), while in Case II (having a lower input rate during low flow: $i_R = 0.3$) some of them survived because of the sediment-starved conditions.

To quantify this effect directly on step formation, we introduce step density $d_S$, defined as the ratio between the number of cross-sections with steps and the total number of cross-section of the channel. The variable $d_S$ can vary between 0 when no steps are present in the channel, and 1 when all the channel morphology is made by steps. The definition of a step is not straightforward, even in the field and in the lab where many different identification algorithms have been proposed (e.g. Milzow et al., 2006; Zimmermann et al., 2008). Since our goal here is not to identify and count the number of steps or to test which step identification algorithm works best, we simply define a step in terms of local departure from the equilibrium channel slope, similarly to the method of Milzow et al. (2006). The steady-state simulations give us the value of the final slope at equilibrium for a given set of parameters ($E$ and $i_R$). We define that a cross-section in $CAST$ has a step if its local slope is greater than the equilibrium slope by a factor $\beta$. The time-series of step density evaluated in this way is shown for different values of $\beta$ in Fig. 14. The temporal pattern of step density variations is largely independent of $\beta$. The time evolution of step density as a function of flow and sediment supply conditions allows us to draw two conclusions. First, there is a clear difference between simulations with and without jamming in that jamming is responsible for step formation, and without it there are practically no steps formed in the channel (blue and red lines in Fig. 14). Second, after steps are generated during high-flow periods due to jamming, they only survive during low flow if the sediment supply decreases (yellow lines in Fig. 14), in sediment-starved conditions. This illustrates the temporal dynamics of step counts as observed in the field (e.g. Molnar et al., 2010) and in flume experiments (e.g. Curran and Wilcock, 2005).

# 6 Discussion

## 6.1 Bed load: a Stochastic, Granular and Shredding Phenomenon

The $CAST$ model without jamming, $CAST_{RBM}$, simulates bed load transport over a rough-bed at the grain scale, considering particle entrainment as a stochastic process driven by a local exposure. Our model describes bed load from a grain prospective because local granular effects in particle mobility and transport are key for developing a bed morphology, especially in steep and well-structured streams. Kirchner et al. (1990) pointed out the role of granular interactions between gravel particles on a river bed and showed how the erodibility of a grain is controlled by its protrusion and friction angle. Given the associated high variability, they also suggested that, instead of using one single value for the shear stress, a probabilistic approach should be applied. The role played by particle interlocking and partial burial in increasing measured friction angles in steep channels has been shown recently also by Prancevic and Lamb (2015). Many laboratory studies have increased our knowledge of bed load transport exactly by looking at the granular scale (e.g. Lajeneusse et al., 2010; Houssais et al., 2015), suggesting that we

might be more successful in describing this phenomenon when borrowing concepts from the granular physics community (e.g. Church and Zimmermann, 2007; Frey and Church, 2011).

$CAST$ assumes a stochastic description of sediment transport, following and corroborating recent research (Furbish et al., 2012; Roseberry et al., 2012; Heyman et al., 2014; Ancey and Heymann, 2014). Our model produces fluctuations in transport rates by the interaction with the bed through entrainment and deposition of individual particles (Ancey and Heymann, 2014) and these fluctuations are observed in our simulations even with a constant input forcing. What in our model is defined as specific sediment flux $q_S$ is equivalent to the particle activity as defined in Furbish et al. (2012). In a companion paper Roseberry et al. (2012) found that changes in transport rates are dominated by changes in the number of particles in motion rather than velocity: this justifies our choice of assuming constant particle velocity but varying entrainment threshold in our simulations.

The stochastic parameterization of $CAST$ does not assume a priori any probability distribution for particle hop distances, and yet they turn out to be well fitted by an exponential distribution, in agreement with previous theoretical and field studies (e.g. Hill et al., 2010; Hassan et al., 2013; Schneider et al., 2014). The fact that, despite its simplicity, the model can reproduce in a robust way this important feature, proves at least partially that the local grain-grain and grain-bed interaction rules in $CAST$ are appropriate and that the phenomenological descriptions of the simulated phenomena are going in the right direction.

Finally, $CAST$ reproduces also the shredding effect sometimes visible in sediment transport (Jerolmack and Paola, 2010). The measured variability of sediment flux and its fluctuations are dictated by the internal dynamics of the system and the degree of fluctuations in the input forcing does not always affect the sediment flux in a clear way (see Fig. 8). Our results then show the RCM potential of modeling of bed load transport as a stochastic phenomenon at the grain scale. Interactions between individual particles can give rise to, or at least strongly impact, the variability observed in natural fluvial systems. Reduced-complexity models like $CAST$ can serve to model these interactions and their effects and can be used to gain new insights into the complex dynamics of sediment transport and to test new research hypotheses.

## 6.2 Step Formation and Stability: a Granular Problem

We showed with $CAST_{JM}$ that dynamic jamming of particles in motion is effective in forming steps (see Fig. 13). In fact only by including the jamming process we did generate step-pool like morphologies in our numerical experiments. Moreover, once steps are formed, they remain stable if the flow conditions change (i.e. the entrainment probability decreases) and the supply of sediment is low enough to avoid that these steps are buried by particles (as shown in Fig. 14). These results are consistent with the main ideas of the jammed state hypothesis of Church and Zimmermann (2007) who theorized and showed experimentally (Zimmermann et al., 2010) that step stability needs (a) jamming, expressed as a low width to diameter ratio so as to enhance granular forces, (b) low flow stage, in order to avoid the mobilization of keystones, and (c) sediment-starved conditions, because a too high sediment concentration would bury the steps. Despite its simplifications, especially uniform sediment and no explicit flow parametrization, $CAST_{JM}$ can reproduce these observations and support the jammed state hypothesis for step stability.

We did not observe any specific wavelength of step occurrence, as usually predicted by hydraulic-based theories on step formation (e.g. Whittaker and Jaeggi, 1982). Given the stochastic nature of $CAST$, steps due to jamming are not formed with a regular spacing. Moreover, as it has been shown by more recent experimental (Curran, 2007; Zimmermann et al., 2010) and

field studies (Zimmermann and Church, 2001; Molnar et al., 2010), step occurrence is mainly driven by the random location of boulders (i.e. keystones) around which sediment deposits and clusters.

## 6.3 Outlook

Our modeling approach has by definition some simplifications and limitations which we think can be improved in future research. First, the uniform size of the sediment prevents us to model specifically any grain-size effect that might indeed be very important in steep-channel dynamics. We partially incorporated these effects in the stochastic parameterization of entrainment: the fact for the same value of relative exposure $R$ and entrainment parameter $E$ some particles are displaced and some are not accounts also for differences in their dimension and weight. Also the jamming process may be grain-size dependent, as well as the particle velocity. Second, the parameterization of changing flow conditions is done indirectly, summarized entirely in the entrainment parameter $E$. This is done mainly because we are not aiming to model discharge, flow, shear stress on the bed, but rather transfer their effects onto the probability of entraining grains. However, future improvements of $CAST$ could include a more direct relation between hydraulic stresses on the bed and the $E$ and $S$ parameters in our model. Third, the granular interactions (i.e. collisions) among particles always lead to deposition, which might not always be realistic, at least in fluvial systems with particles having different sizes and shapes. The same can be said about interactions with the banks of the channel. However, in a uniform-size case this assumption does not seem to be too strong. Furthermore, we did not account for the transfer of momentum that could happen when a particle is deposited and so enhancing the probability of entrainment of the surrounding grains (i.e. "collective entrainment" as in Ancey and Heymann (2014)). With a model like $CAST$ the relative importance of this phenomenon in the entrainment process could be evaluated. Finally, the choice of representing the jamming of grains on the bed as a permanent process is a limitation. In a model having only a single grain size, this choice has been made to account for the additional granular forces that are making step structures more stable around a keystone. In future research, especially in a model which accounts for different grain fractions, the role of step formation and stability may be transferred to the coarsest grains to which jamming threshold will apply.

In conclusion, in our opinion the strongest limitation of the current model is the absence of sediment-sorting and other grain-size effects. All these phenomena will be incorporated in the next version of the model which will have different grain-size fractions.

## 7 Conclusions

We presented a new particle-based reduced-complexity model $CAST$ (Cellular Automaton Sediment Transport) that simulates bed load transport and changes in channel morphology including the processes of jamming and step formation. The model simulates grain-grain and grain-bed interactions with uniform-size particles and can have stochastic or deterministic parameterizations for sediment input rate and particle entrainment. With only few parameters, it it possible to simulate channels with different sediment supply and flow conditions. At steady state, $CAST$ can reproduce a realistic bed morphology and typical fluctuations in transport rates, whose memory features are consistent with previous experimental data. Moreover, particle hop

distances are well-fitted by exponential distributions, in agreement with field observations. One of the main results is the role played by stochasticity both in the entrainment and in the input rate. A stochastic input rate does not change the final outcome of the model compared to a constant input having the same mean. However, if the entrainment is modeled deterministically, the resulting channel does not have the typical variable bed roughness encountered in real fluvial systems.

5      The dynamical effect of particle jamming was added to test under which conditions steps are formed and remain stable in steep channels. The effect of jamming has been tested in unsteady simulations where the entrainment probability and the input rate have been changed to simulate a sequence of high-flow and low-flow periods. $CAST$ generates step structures during high-flow periods that survive during low flows in simulations with sediment-starved conditions, in agreement with the jammed-state hypothesis. Our results support the jammed-state hypothesis as a framework to explain step formation and stability and, more

10    in general, they show the potential of reduced complexity-models at a grain scale with stochastic parameterizations. We are of the opinion that models such as $CAST$ can give new insights into the dynamics of complex phenomena like sediment transport and step formation and be useful to test research hypotheses in fluvial geomorphology.

*Acknowledgements.* Matteo Saletti acknowledges the support of the SNSF grant number 200021_140488 that funded his PhD and of the SNSF Doc.Mobility grant that allowed him to spent a semester at the Geography Department of UBC in Vancouver (Canada). Fruitful

15    conversations with Chris Paola and Shawn Chartrand are greatly acknowledged, as well as many inputs received during the workshop 'Complexity in geomorphology' held at ETH in June 2015. Prof. Chris Paola and one anonymous referee provided helpful comments, which helped us to improve the manuscript. Finally, the authors wish to acknowledge the ETH cluster EULER, where the numerical simulations have been performed with the help of Dr. Daniela Anghileri.

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

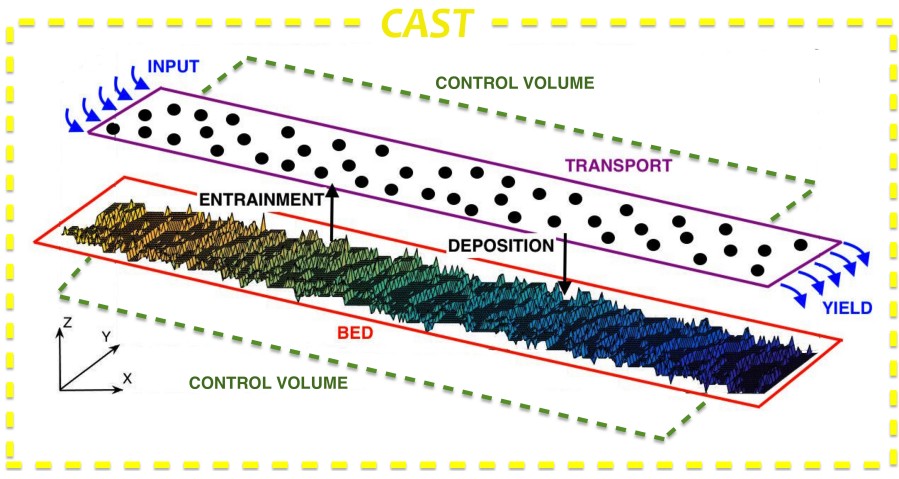

**Figure 1.** Sketch of the model. The space is discretized in a longitudinal dimension $X$ and a transversal dimension $Y$. Bed elevation is given by the coordinate $Z$. Particles can be either in the bed matrix or in the transport matrix. They can enter the transport matrix as sediment input from the upper boundary or by entrainment from the bed, while they can leave the transport matrix as sediment output or by deposition on the bed. Sediment is input at the upstream boundary and simulated as sediment yield leaving the downstream boundary.

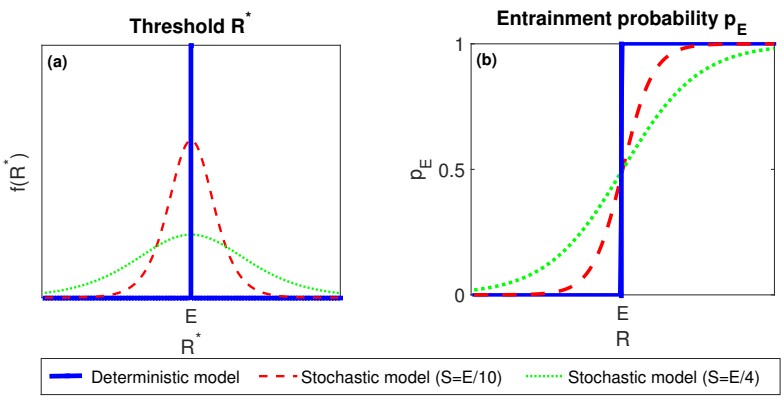

**Figure 2.** Deterministic and stochastic parameterization of entrainment in $CAST$. (a) The probability density function of the threshold $R^*$. (b) Entrainment probability as a function of relative exposure $R$.

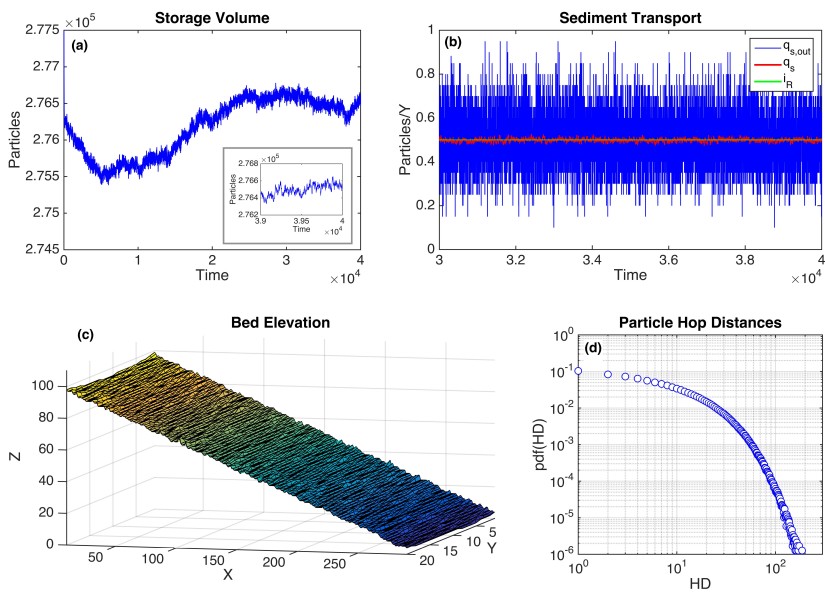

**Figure 3.** $CAST_{RBM}$ steady-state results of simulation with $E = 1.5$ and $i_R = 0.5$. (a) Time series of storage volume: adjustment phases of aggradation and degradation around the equilibrium condition. In the small box a zoom in the last 1000 time steps is shown. (b) Time series of sediment transport: even with a constant input rate (green line) the sediment flux fluctuates. Fluctuations are large if measured in a single cross section at the downstream end (blue line), but become smaller if averaged over the entire control volume of the channel (red line). (c) Bed elevation in the final configuration: the model produces a rough bed with particles having different exposures $R$. (d) Probability density function of particle hop distances with an exponential distribution estimated over more than 2 million simulated particle paths.

**Table 1.** Values of the parameters used in the steady-state simulations

| Parameter | Name | Value(s) |
|---|---|---|
| Channel length | $X$ | 300 d |
| Channel width | $Y$ | 20 d |
| Simulation duration | $T$ | 40000 |
| Deposition parameter | $D$ | 0.5 |
| Shape parameter | $S$ | $\frac{E}{5}$ |
| Entrainment parameter | $E$ | $1 \div 2$ |
| Specific input rate | $i_R$ | $0.2 \div 0.7$ |

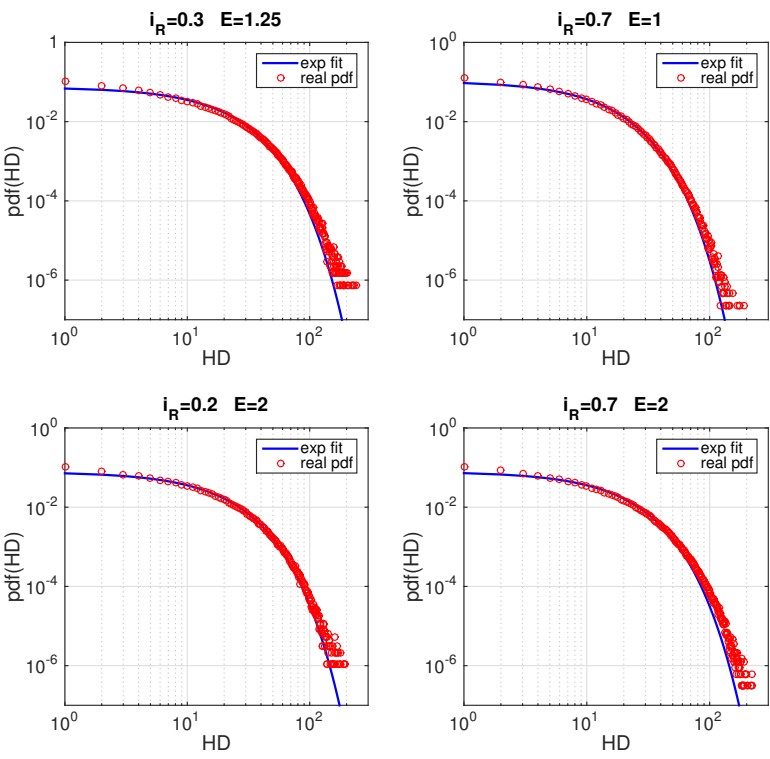

**Figure 4.** Probability density functions of simulated particles hop distances (red dots) fitted with an exponential distribution (blue line) for 4 different parameter sets.

**Table 2.** Values of the parameters used in the unsteady simulations.

| Parameter | Name | Case I | Case II |
|---|---|---|---|
| Channel length | $X$ | 300 d | 300 d |
| Channel width | $Y$ | 20 d | 20 d |
| Simulation duration | $T$ | 550000 | 550000 |
| Low flow duration | $T_{low}$ | 70000 | 70000 |
| High flow duration | $T_{high}$ | 70000 | 70000 |
| Rising limb duration | $T_{ris}$ | 2000 | 2000 |
| Falling limb duration | $T_{fal}$ | 8000 | 8000 |
| Entrainment parameter (low flow) | $E_{low}$ | 2 | 2 |
| Entrainment parameter (high flow) | $E_{high}$ | 1 | 1 |
| Specific input rate (low flow) | $i_{R,low}$ | 0.4 | 0.3 |
| Specific input rate (high flow) | $i_{R,high}$ | 0.4 | 0.6 |

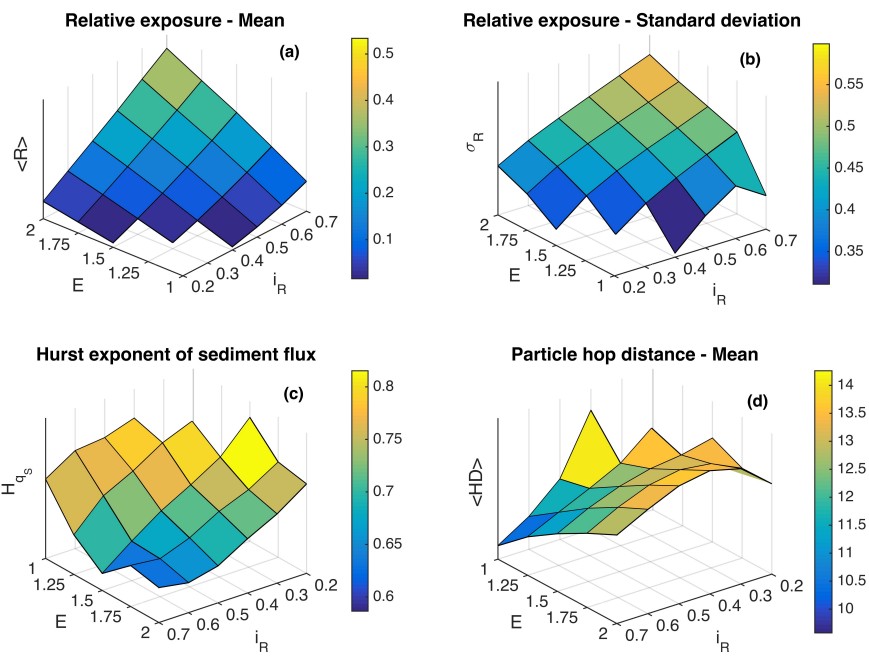

**Figure 5.** Selected $CAST_{RBM}$ variables as a function of input rate and entrainment: (a) Mean relative exposure $<R>$, (b) Standard deviation of relative exposure $\sigma_R$, (c) Hurst exponent of the specific sediment flux $H_{q_S}$, and (d) Mean particle hop distance $<HD>$.

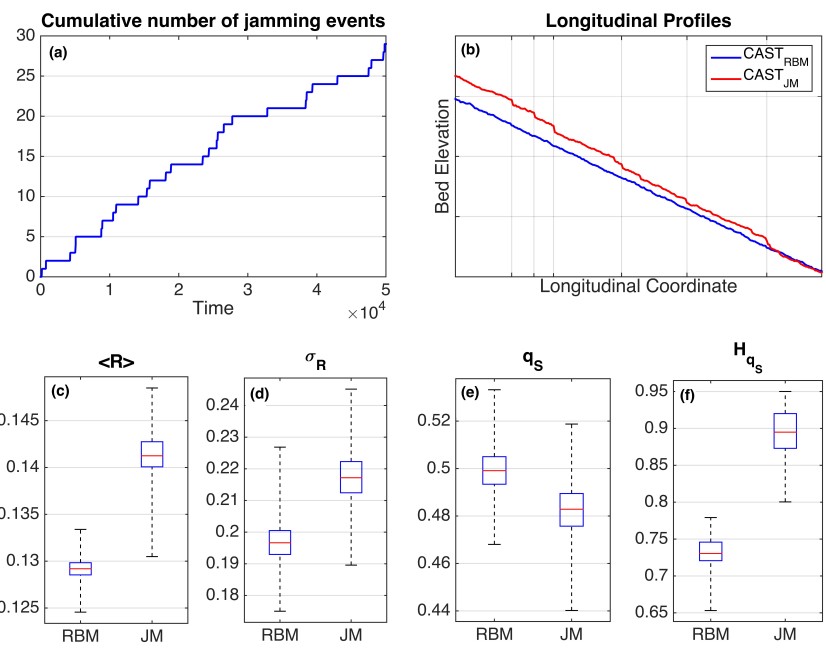

**Figure 6.** Comparison between simulations without jamming (RBM) and with jamming (JM) for $i_R = 0.5$ and $E = 1.25$. (a) Cumulative number of jammed cross-sections. (b) Longitudinal profiles at the end of the simulations. (c) Box-plots of the instantaneous values of mean relative exposure. (d) Boxplots of the instantaneous values of standard deviation of relative exposure. (e) Boxplots of the instantaneous values of specific sediment flux. (f) Box-plots of the values of Hurst exponent of specific sediment flux computed from the 20 realizations.

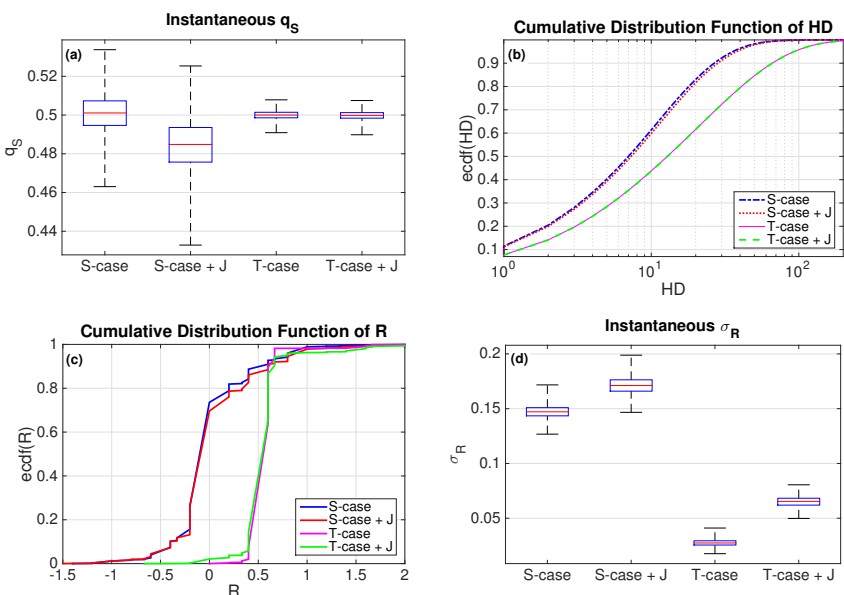

**Figure 7.** Comparison between stochastic entrainment ($S-case$ without jamming and $S-case+J$ with jamming) and threshold entrainment ($T-case$ without jamming and $T-case+J$ with jamming). (a) Boxplots of the instantaneous values of specific sediment flux $q_S$. (b) Empirical cumulative distribution function of particle hop distances $HD$. (c) Empirical cumulative distribution function of relative exposure $R$ computed on the entire control volume at the end of the simulation. (d) Boxplots of the instantaneous values of the spatial standard deviation of relative exposure $\sigma_R$.

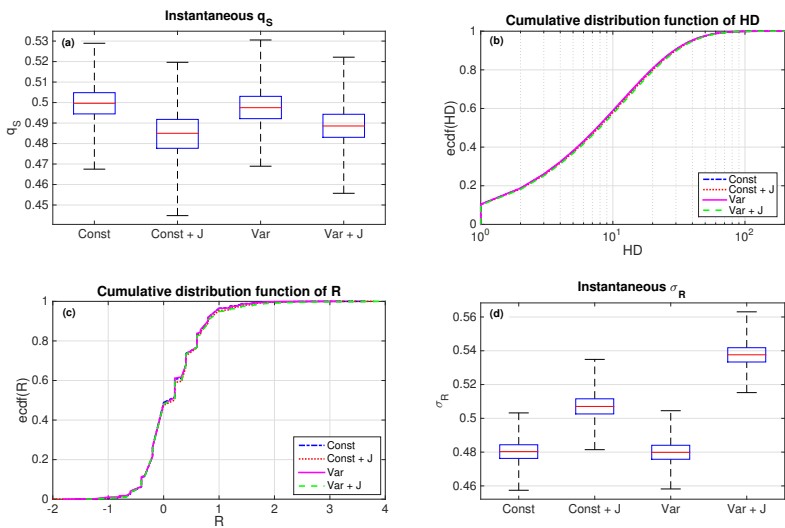

**Figure 8.** Comparison between constant input rate ('Const' without jamming and 'Const + J' with jamming) and variable input rate ('Var' without jamming and 'Var + J' with jamming). (a) Boxplots of the instantaneous values of specific sediment flux $q_S$. (b) Empirical cumulative distribution function of particle hop distances $HD$. (c) Empirical cumulative distribution function of relative exposure $R$ computed on the entire control volume at the end of the simulation. (d) Boxplots of the instantaneous values of the spatial standard deviation of relative exposure $\sigma_R$.

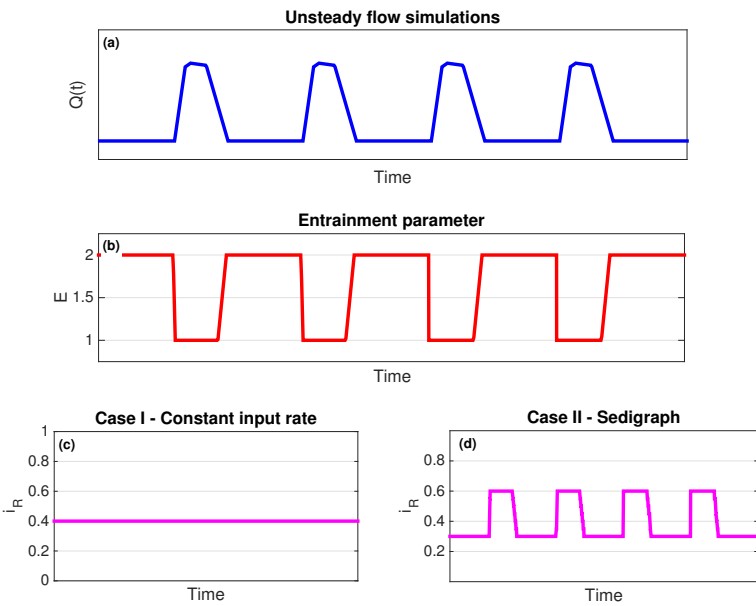

**Figure 9.** Unsteady simulations with 4 consecutive floods. (a) Generic hydrograph the model is simulating. (b) Variation of the entrainment parameter $E$ to simulate the changing flow conditions. (c) Case I: simulations with varying $E$ and constant input rate $i_R$. (d) Case II: simulations with varying $E$ and varying input rate $i_R$.

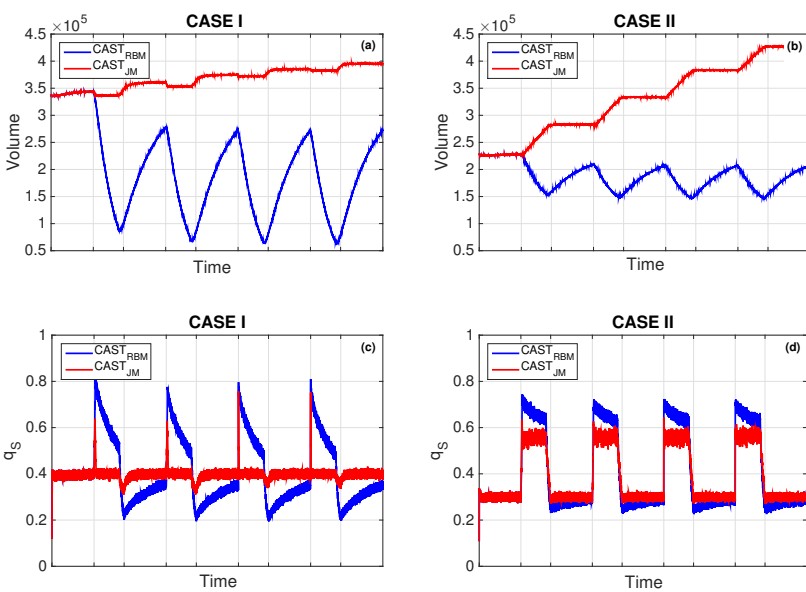

**Figure 10.** Unsteady simulations with 4 consecutive floods. (a) Time series of storage volume for the constant input case. (b) Time series of storage volume for the variable input case. (c) Time series of specific sediment flux for the constant input case. (d) Time series of specific sediment flux for the variable input case.

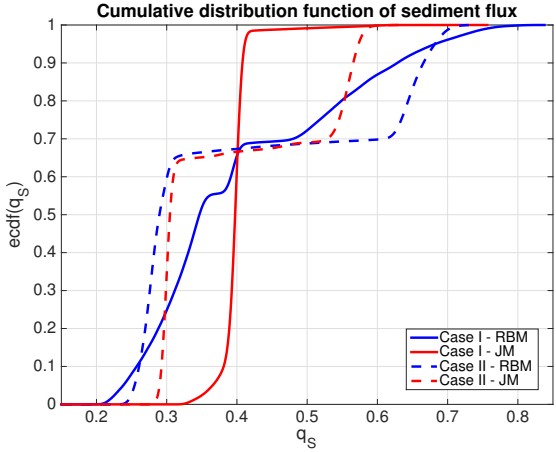

**Figure 11.** Unsteady simulations with 4 consecutive floods. Empirical cumulative distribution functions of specific sediment flux. Blue identify the rough-bed model, red the jamming model. Solid lines identify the constant input case (Case I), dashed lines the variable input case (Case II).

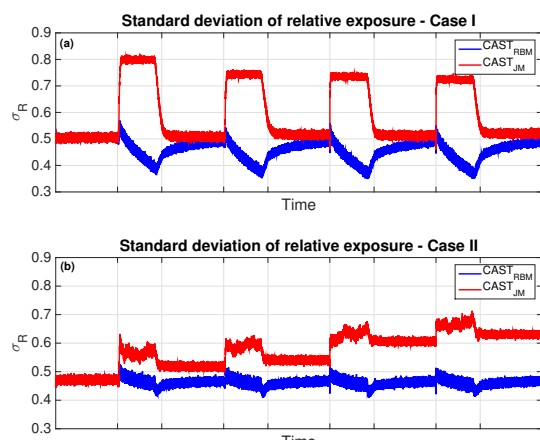

**Figure 12.** Unsteady simulations with 4 consecutive floods. Time series of the standard deviation of relative exposure $\sigma_R$ for (a) Case I and (b) Case II, both for the rough-bed case (in blue) and the jamming case (in red).

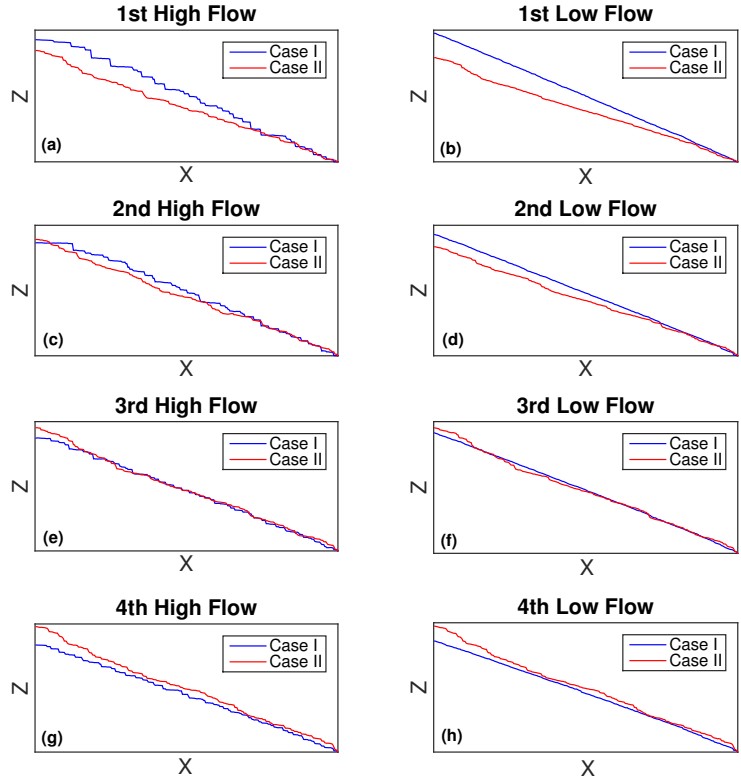

**Figure 13.** Unsteady simulations with 4 consecutive floods. Longitudinal profiles of bed elevation computed at the end of every high-flow period (left column: a, c, e and g) and at the end of each of the following low-flow period (right column: b, d, f and h) for a typical simulation.

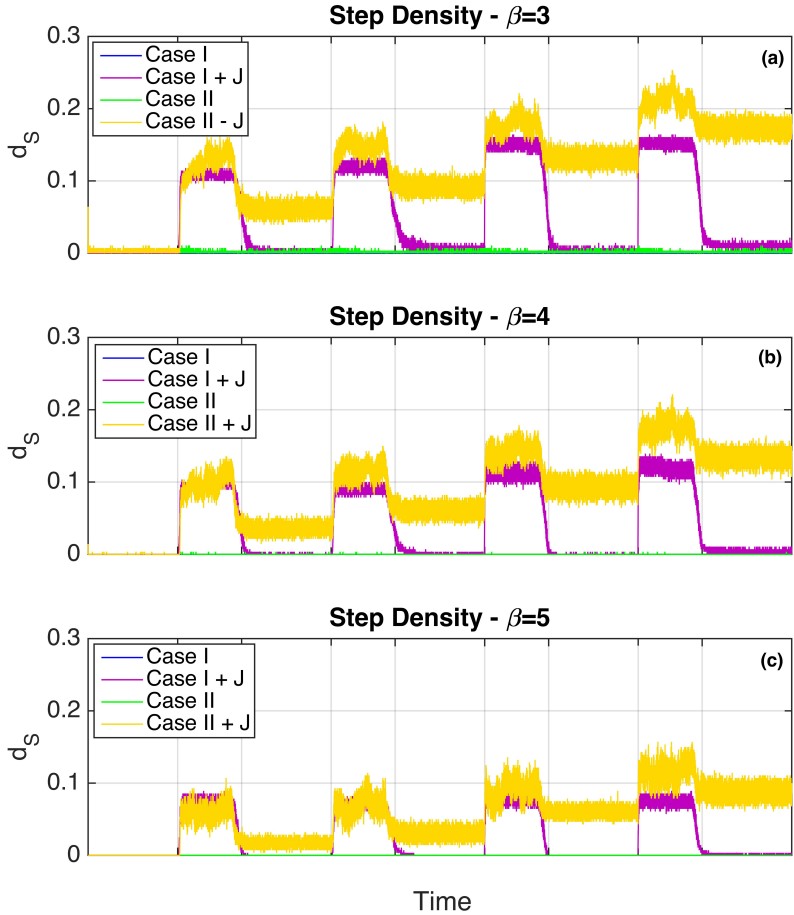

**Figure 14.** Unsteady simulations with 4 consecutive floods. Time series of step density with (a) $\beta = 3$, (b) $\beta = 4$ and (c) $\beta = 5$.