# Peer review of "A reduced-complexity model for sediment transport and step-pool morphology"

_Earth Surface Dynamics, 2016_

## Referee Comment (RC1) · Anonymous Referee #1 · 26 Apr 2016

This manuscript presents a new model for sediment transport in steep streams using a stochastic cellular automaton model. It specifically addresses the formation of step-pools incorporating into the model a so-called jamming process. The paper is well-written and gives a nice overview of an intensive field of research at the interface between geomorphology and the physics of sediment transport. However, there are a few major issues that need to be addressed and I consider that this manuscript needs to be significantly modified to be useful for a large number of researchers in these communities. I hope that the following comments will help.

[Figure]

Major issues

- A roughness is a measure of amplitude. It seems very awkward to call *roughness* a variable that can be negative. What is called the *roughness* in this manuscript is basically a local downstream slope.

- This is not because you develop a "*Reduced Complexity Model*" (RCM) that it is not necessary to check some basic relations, such as the relation between sediment transport and slope. But, taking into account the previous comment, this is basically what you do when you investigate the mean roughness with respect to $E$ and $i_{\mathrm{R}}$. This relation should be used to set-up the model and not presented as a result.

- There is no deposition length in the model and you never discuss the characteristic length scale of the exponential decays for particle hop distance. You need to test the dependency of this characteristic length scale to the system length and to describe how it varies with respect to the model parameter values.

- You propose a two-dimensional RCM with only $3 \times 10^3$ cells. This is two orders of magnitude smaller than the actual number of particles in continuous numerical models that solve turbulent flow and particle collisions. I do understand that size does not matter but you should explain and justify why such a small section of the bed is enough in your model.

- Please clarify your initial condition. This is particularly important because I have the feeling that you can have stationary states for which there is no erosion or deposition. Furthermore, the steady-state is not very convincing from the fluctuations observed in Fig. 3a.
  You should also be more careful about your downstream boundary conditions and explain how the 10 sections that are removed affect the results.

Jamming

- Is there an increase of the sediment flux downstream leading to saturation and then to jamming? If it happens to be the case, how does this mechanism relate to the saturation flux and the saturation length in classical transport model?

- In relation with the previous point, is there a characteristic wavelength for the formation of step-pools? It could be informative to characterize the height distribution of these step-pools.

- Before exploring the role of variable flow strength, it is necessary to understand the dynamics of step-pool formation. I have the feeling that the huge amount of deposition create a barrier, which is subsequently responsible for net deposition upstream. Is it the case in the model and in nature?

- I suggest to plot a space-time diagram for the formation (and the disappearance) of step-pools.

---

## Author Comment (AC1) · 2 May 2016

**REPLY TO THE COMMENTS OF REFEREE #1**

**by Saletti Matteo et al.**

We would like to thank the Referee for his/her helpful comments. Here is how we intend to address the issues that have been raised:

*A roughness is a measure of amplitude. It seems very awkward to call roughness a variable that can be negative. What is called the roughness in this manuscript is basically a local downstream slope.*

> We realize that roughness is not a good lexical choice and we indeed had discussed other alternatives. In the revised manuscript we have decided to call this variable Ex (Exposure).

*This is not because you develop a "Reduced Complexity Model" (RCM) that it is not necessary to check some basic relations, such as the relation between sediment transport and slope. But, taking into account the previous comment, this is basically what you do when you investigate the mean roughness with respect to E and $i_R$. This relation should be used to set-up the model and not presented as a result.*

> We plan to modify what is now in the first part of the results, related to the steady-state simulations, in the direction suggested by the referee. We plan to add a new section (e.g. 'Model Set-Up') where we will use some of the outcomes of the steady-state simulations to show the basic relations suggested by the referee.

*There is no deposition length in the model and you never discuss the characteristic length scale of the exponential decays for particle hop distance. You need to test the dependency of this characteristic length scale to the system length and to describe how it varies with respect to the model parameter values.*

> The referee is correct in saying that we do not specify a-priori any deposition length: the exponential distribution of particle hop distances arises as a consequence of the model's rules. In Figure 5d we show how the mean HD varies as a function of the entrainment parameter E and the input rate $i_R$. We do not discuss it in relation to the system length X because the maximum value of particle HD is always much smaller than X. In the revised manuscript we plan to give more relevance to this part and better discuss the implications of our assumptions.

*You propose a two-dimensional RCM with only $3{\times}10^3$ cells. This is two orders of magnitude smaller than the actual number of particles in continuous numerical models that solve*

*turbulent flow and particle collisions. I do understand that size does not matter but you should explain and justify why such a small section of the bed is enough in your model.*

We realize that the "scale-issue" has not been addressed directly in the manuscript. We plan to do that in the revised version, by running a simulation with much greater grid dimensions, in order to show that size does not have an effect on the results, but only on the time needed to reach the equilibrium point.

*Please clarify your initial condition. This is particularly important because I have the feeling that you can have stationary states for which there is no erosion or deposition. Furthermore, the steady-state is not very convincing from the fluctuations observed in Fig. 3a. You should also be more careful about your downstream boundary conditions and explain how the 10 sections that are removed affect the results.*

In the simulations having a constant set of parameters (i.e. steady-state case) the initial conditions (ICs) of the system do not matter: the system is going to reach always the same "equilibrium point" in terms of average fluxes in and out of the reach. ICs only determine how long it will take to reach this attractor point. Fig. 3a, which is a zoom in a restricted time window, aims to show this is a rather dynamic equilibrium point, around which the system fluctuates. We realize this is an important point that will be discussed in more detail in the revised manuscript. The downstream boundary condition is not influencing the system already a few cross-sections upstream. We removed 10 cross-sections from the control volume just to be sure to avoid any kind of influence as is normally done in simulation.

*Is there an increase of the sediment flux downstream leading to saturation and then to jamming? If it happens to be the case, how does this mechanism relate to the saturation flux and the saturation length in classical transport model?*

Dynamic jamming is actually due to a sort of saturation process in the transport layer (i.e. it happens when the transport layer is full with moving particles). However, this process in the model is localized in single cross-sections, and does not have a direct relation to saturation length theories. Jamming is indeed caused by a local increase in sediment flux, which may be due to global changes in flow conditions (i.e. increase of the entrainment probability) or stochastic fluctuations in transport rates.

*In relation with the previous point, is there a characteristic wavelength for the formation of step-pools? It could be informative to characterize the height distribution of these step-pools.*

We did not address directly step-pool geometry and statistics (such as wavelength, height, spacing, etc), focusing only on step formation and stability as a function of jamming and sediment supply. We did not observe any specific step spacing, wavelength or step height that stand out beyond what would be expected from purely geometric considerations. However, we will look at some of these aspects in the revision process.

*Before exploring the role of variable flow strength, it is necessary to understand the dynamics of step-pool formation. I have the feeling that the huge amount of deposition creates a barrier, which is subsequently responsible for net deposition upstream. Is it the case in the model and in nature?*

> The referee is correct: the process of jamming enhances deposition upstream as it happens in nature where steps are formed, among other factors, by deposition and clustering of bed material around large clasts (i.e. keystones). Of course in nature the process is much more complex (depending on hydraulic conditions, geometry, grain-size, etc): here we try to address the step-formation process with a RCM approach by assuming that the granular effect of jamming is what drives step formation. There are other ways of forming steps, e.g. by erosion of the bed around keystone clasts, but we are of the opinion that flume experiments and especially steps in natural streams form predominantly by the joint blocking of coarse grains in motion close to threshold conditions.

*I suggest to plot a space-time diagram for the formation (and the disappearance) of step-pools.*

> A diagram for step formation and stability (like the jammed-state diagram of Church & Zimmermann [2007]) would be a nice addition to the paper. We plan to work on that in the revised version.

---

## Referee Comment (RC2) · Christopher Paola (Referee) · 20 May 2016

The paper presents a new reduced-complexity model for the evolution of bed topography and particle motion in steep, coarse-bedded streams, with a focus on step-pool topography. The model includes only one particle size, which is capable of only the simplest of stepwise motions on a rectangular grid. Entrainment and deposition are controlled by particle protrusion as measured by height relative to the average of the surrounding bed, with entrainment modeled in a somewhat more detailed way than deposition, which occurs simply when a particle lands on a location lower than its average surroundings. Entrainment can be either deterministic when protrusion exceeds a fixed threshold, or probabilistic. The only other major ingredient in the model is the (optional) inclusion of a jamming condition, which again is the simplest possible one: the chance arrival of particles across a whole channel cross-section, leading to imme-

diate stoppage. There is no explicit role for the flow, which appears only indirectly via the entrainment threshold.

In my opinion, this is a true "Saint Exupery" model: there is nothing left to take away. The authors have done exactly what the first formation of a model for a self-organized pattern like step-pool topography should do: to find out the absolute minimum set of conditions necessary to produce it. With only simple particle interactions and the crudest possible representation of the flow, under the right conditions the model evolves what looks to me to be pretty convincing step-pool topography. I enthusiastically endorse this for publication. It is a beautiful example of reduced-complexity modeling that sheds new light on how this important feature of steeplands streams, important both in terms of sediment transport and stream ecology (e.g. the pools are important shelter zones for fish), develops and responds to changes in sediment supply and water discharge. It will attract the interest of the large community that studies mountain rivers, as well as the growing community of people interested in applying granular physics (whence the basic insight about jamming) to sediment transport. It will also inspire further research, some of which will have the unfortunate aim of adding detail in the interest of making the model "more realistic". Nonetheless, this paper will stand as the one that revealed the essence of the phenomenon.

Comments: 1. One thing that is left to the reader to infer is the relation between particle entrainment/deposition and changes in bed topography. I assume this is as simple as I imagine: that when a particle is deposited, the elevation ($Z_{ij}$ in the paper) is incremented by a unit amount, and vice versa for erosion. But this should be stated explicitly.

2. The authors' choice of probabilities for a particle moving straight downstream as opposed to stepping left or right (2.2.2) seem a bit arbitrary. Are there any observations of particle paths in steep streams that could be used to constrain these? How much does the choice of weights matter for the observed model outcomes?

3. The authors state (2.3) that, in the jamming model, once the jamming condition is met, all the particles across the jammed section are 'locked' in place. A little later the text says the process is 'permanent'. I took this to mean that they were deposited and could never move again, but reading on and thinking about the results reported later, I don't see how this could be right. This is an important point, since the inclusion of jamming is an important part of the paper, so it must be explained clearly. I think what the authors mean is that all the particles in that section are considered to be deposited, i.e. stop moving, regardless of the local relative elevation. But then, I assume, the jammed particles can be entrained again according to the same criterion used for all the other particles, i.e. there is nothing special about particles that were deposited through jamming. Whether this is correct or not, the authors should explain clearly what if any conditions are needed to re-entrain particles deposited through the jamming criterion.

4. In section 2.2.2, it appears that deposition occurs if a particle in motion collides with another particle in transport. This seems inconsistent with the original simple condition for deposition, which is simply that the particle arrives in a pocket (relative elevation deficit). This should be clarified. Also, when there is a collision between two moving particles, are both deposited, or only one? It would also be interesting, and in my view still in keeping with the aim of a simplest-possible model, to see whether changing the entrainment condition to account for particle collision (e.g. by letting the entrainment probability increase upon arrival of a moving particle) would change the model behavior.

---

## Author Response (AR1)

Zurich, June 10th 2016

Dear Prof. Castelltort,

We would like to thank both Referees for the time they took to review our manuscript [esurf-2016-15], providing interesting and helpful comments. Thanks to that, in our opinion the manuscript we are resubmitting clarifies many points that were not clear in the first version and both structure and terminology have been modified to avoid misunderstandings.

We performed a review of our previous manuscript addressing the points raised by the two referees, clarifying the unclear points, changing partially the structure and the presentation and improving some figures (mainly Fig.1 and Fig. 3); we also attach a 2-page pdf file as Supplementary Material. Moreover, we changed sentence structures to help readability and improved the reference list.

In the next pages you may find the single points raised by the two Referees (in italics) followed by our reply and the way in which the manuscript has been modified to address the issues that have been raised.

Finally, you may find the new version of the manuscript, with all changes highlighted in yellow.

Best regards,

Matteo Saletti
* * *
Matteo Saletti, M.Sc.
PhD Candidate

ETH Zürich
Institute of Environmental Engineering (IfU)
Hydrology and Water Resources Management
HIF CO 46.5 - Stefano-Franscini-Platz 3
8093 - Zurich - Switzerland

Phone  : +41 (0) 44 633 33 84
E-mail  : saletti@ifu.baug.ethz.ch

**RESPONSE TO REFEREE #1**

*A roughness is a measure of amplitude. It seems very awkward to call roughness a variable that can be negative. What is called the roughness in this manuscript is basically a local downstream slope.*

> We realize that roughness was not a good lexical choice and we indeed had discussed other alternatives. In the revised manuscript we have decided to call this variable relative exposure.

*This is not because you develop a "Reduced Complexity Model" (RCM) that it is not necessary to check some basic relations, such as the relation between sediment transport and slope. But, taking into account the previous comment, this is basically what you do when you investigate the mean roughness with respect to E and $i_R$. This relation should be used to set-up the model and not presented as a result.*

> We agree with the Referee that some of the outcomes of the model that were presented in the Results section (i.e. the steady-state simulations) in the original manuscript are actually part of the model set-up. In the revised manuscript we present them in the section "Model Set-Up" (Section 3), separated from the results shown in the following sections.

*There is no deposition length in the model and you never discuss the characteristic length scale of the exponential decays for particle hop distance. You need to test the dependency of this characteristic length scale to the system length and to describe how it varies with respect to the model parameter values.*

> As correctly stated by the Referee we do not specify a-priori any deposition length: the exponential distribution of particle hop distances arises as a consequence of the model's rules. The values of HD obtained in the simulation do not depend on the scale of the system: the maximum value of HD is always much lower than the system length X. This is now stated clearly at the end of Section 3.1.1 (P9 L9-11). The dependency of HD on the model parameters is shown in Figure 5d and discussed at the end of section 3.2 (P10 L6-11): here we explain the changing values of the mean hop distances as a consequence of both particle-particle interactions and particle-bed interactions.

*You propose a two-dimensional RCM with only 3×103 cells. This is two orders of magnitude smaller than the actual number of particles in continuous numerical models that solve turbulent flow and particle collisions. I do understand that size does not matter but you should explain and justify why such a small section of the bed is enough in your model.*

> We realize that the "scale-issue" has not been addressed directly in the previous version manuscript. In the revised version of the manuscript, we now address the effect of scale at the beginning of section 3 (P8 L3-6), explaining why it does not matter for the purpose of our study. Moreover we have run simulations with a grid up to two orders of magnitude longer, in order to show that the system is getting to the same equilibrium slope and displays the same trends no matter the size. Scale has only an influence on the time required to reach this equilibrium points. We report these findings now in the Supplementary Material.

*Please clarify your initial condition. This is particularly important because I have the feeling that you can have stationary states for which there is no erosion or deposition. Furthermore, the steady-state is not very convincing from the fluctuations observed in Fig. 3a. You should also be more careful about your downstream boundary conditions and explain how the 10 sections that are removed affect the results.*

In our model the initial conditions (ICs) of the system do not influence: the system is going to reach always the same "equilibrium point" in terms of average fluxes in and out of the reach. ICs only determine how long it will take to reach this attractor point. As ICs we specify a starting slope and then we perturb the channel with some random noise. We now specify that clearly in Section 2.1.5 (P7 L8-11).

By "steady state" we did not mean a state in which nothing is changing, but rather a point at which the mean sediment fluxes in and out of the channel are equal, and the storage volume fluctuates around an equilibrium value. In Fig. 3a, we know show two time-series of the storage volume, without and with a zoom in a restricted time window: we hope this will help to clarify that the equilibrium point is a dynamic one, around which the system fluctuates, as also explained in Section 3.1 (P8 L22-25).

The downstream boundary condition is not influencing the system already a few cross-sections upstream. We removed 10 cross-sections from the control volume just to be sure to avoid any kind of influence as is normally done in simulation.

*Is there an increase of the sediment flux downstream leading to saturation and then to jamming? If it happens to be the case, how does this mechanism relate to the saturation flux and the saturation length in classical transport model?*

Dynamic jamming is actually due to a sort of saturation process in the transport layer (i.e. it happens when the transport layer is full with moving particles). However, this process in the model is localized in single cross-sections, and it does not relate to saturation length theories. Jamming is indeed caused by a local increase in sediment flux, which may be due to global changes in flow conditions (i.e. increase of the entrainment probability) or stochastic fluctuations in transport rates.

*In relation with the previous point, is there a characteristic wavelength for the formation of step-pools? It could be informative to characterize the height distribution of these step-pools.*

We did not address directly step-pool geometry and statistics (such as wavelength, height, spacing, etc), focusing only on step formation and stability as a function of jamming and sediment supply. We did not observe any specific step spacing, wavelength or step height that stand out beyond what would be expected from purely geometric considerations. This is because in our model step formation, due to jamming, is parameterized in a stochastic way and so steps can be generated both very close and very far to each others, depending on the local sediment flux, which may or may not exceed the jamming threshold. The existence of a characteristics wavelength for the formation of steps has been advocated in the past by the "hydraulic theories", in which steps were supposed to be generated as a consequence of antidunes migration (Whittaker and Jaeggi [1982] and review by Church and Zimmermann [2007]); this mechanism for step formation has been shown in more recent studies (e.g. Curran [2007], Zimmermann et al. [2010]) to be far less important than the "keystone" or "jamming" mechanism, in which steps are formed as a

consequence of grain deposition and clustering around a large particle. Nevertheless, a "mean wavelength" can be inferred by looking at the step density variable $d_s$ (e.g. Fig. 14): $1/d_s$ is in fact the mean distance between cross-sections with steps. However, this quantity is not necessarily informative on the real distance, since it might be the case the more than one consecutive cross-section belongs to the same step. In the revised manuscript, in the last paragraph of the discussion in Section 6.2 (P15 L32-34 and P16 L1-2), we now discuss this aspect.

Step height has been shown in the field and in laboratory experiments (see review by Church and Zimmermann [2007]) to scale with the dimension of the largest grain: therefore, in our uniform grain-size model, this is not an informative variable, being always in the interval between 1-2 grains.

*Before exploring the role of variable flow strength, it is necessary to understand the dynamics of step-pool formation. I have the feeling that the huge amount of deposition creates a barrier, which is subsequently responsible for net deposition upstream. Is it the case in the model and in nature?*

The Referee is correct: the process of jamming enhances deposition upstream as it happens in nature where steps are formed, among other factors, by deposition and clustering of bed material around large grains (i.e. keystones). Of course in nature the process is much more complex (depending on hydraulic conditions, geometry, grain-size, etc): here we try to address the step-formation process with a RC approach by assuming that the granular effect of jamming is what drives step formation. There are other ways of forming steps, but we are of the opinion that flume experiments and especially steps in natural streams form predominantly by the joint blocking of coarse grains in motion close to threshold conditions (e.g. Curran [2007], Zimmermann et al. [2010]). In the revised manuscript (in Section 3.3., P10 L32 and P11 L1-4) we now explain more clearly how steps are formed due to jamming in our simulations.

*I suggest to plot a space-time diagram for the formation (and the disappearance) of step-pools.*

For the reasons explained in the reply to the previous comments, the space distribution of steps is not a meaningful aspect, given the assumptions of our model leading to random distributions of steps in the channel. The time-evolution of step formation and disappearance, as a function of flow and sediment supply conditions, is exactly what we plot in Figure 14 by means of the variable step density: this shows the formation of steps during high-flows when the model accounts for dynamic jamming, and their disappearance during low flows, due to deposition and burial, when sediment supply is too high.

**RESPONSE TO REFEREE #2**

*One thing that is left to the reader to infer is the relation between particle entrainment/deposition and changes in bed topography. I assume this is as simple as I imagine: that when a particle is deposited, the elevation ($Z_{ij}$ in the paper) is incremented by a unit amount, and vice versa for erosion. But this should be stated explicitly.*

> This is correct: erosion/deposition change the local topography by reducing/incrementing the local elevation $Z_{i,j}$. This is now stated clearly in the revised manuscript, and explained with Equation 1 (Section 2, P4 L1-3).

*The authors' choice of probabilities for a particle moving straight downstream as opposed to stepping left or right (2.2.2) seem a bit arbitrary. Are there any observations of particle paths in steep streams that could be used to constrain these? How much does the choice of weights matter for the observed model outcomes?*

> Our choice of probability for lateral displacement has been guided by the assumption that the straight direction of movement must be prevalent but at the same time we did want to consider the (small) chance for lateral movements as well. We are not aware of any study able to help us to constrain this value: therefore, we have tested different values in what we considered a reasonable range (i.e. probability of lateral displacements < 20%) and observed that below roughly 10% the final outcome of the model does not change. A further increase in this value has the effect of enhancing deposition (because of more particle-particle collisions) and so to increase the equilibrium slope for a given set of parameters. We have added a sentence to clarify this point in the revised manuscript (section 2.1.2, P5 L4-8).

*The authors state (2.3) that, in the jamming model, once the jamming condition is met, all the particles across the jammed section are 'locked' in place. A little later the text says the process is 'permanent'. I took this to mean that they were deposited and could never move again, but reading on and thinking about the results reported later, I don't see how this could be right. This is an important point, since the inclusion of jamming is an important part of the paper, so it must be explained clearly. I think what the authors mean is that all the particles in that section are considered to be deposited, i.e. stop moving, regardless of the local relative elevation. But then, I assume, the jammed particles can be entrained again according to the same criterion used for all the other particles, i.e. there is nothing special about particles that were deposited through jamming. Whether this is correct or not, the authors should explain clearly what if any conditions are needed to re-entrain particles deposited through the jamming criterion.*

> We agree with the Referee that this is a crucial point in our analyses that needs to be clarified. The process of jamming leads to a permanent deposition without any further possibility of re-entrainment in a given model run. These grains become keystones in the context of Church and Zimmermann (2007). This aims to represent, for sure in an extreme way, the increased stability of step structures given by granular forces (as implied in the jammed-state hypothesis). However, in the same cross-section entrainment and deposition are still possible, except for grains that have been deposited after a jamming event. In the revised manuscript we explain this more

clearly in Section 2.2 (P7 L15-22), also discussing the implications/limitations of such a choice in the Outlook (Section 6.3 P16 L18-22).

*In section 2.2.2, it appears that deposition occurs if a particle in motion collides with another particle in transport. This seems inconsistent with the original simple condition for deposition, which is simply that the particle arrives in a pocket (relative elevation deficit). This should be clarified. Also, when there is a collision between two moving particles, are both deposited, or only one? It would also be interesting, and in my view still in keeping with the aim of a simplest-possible model, to see whether changing the entrainment condition to account for particle collision (e.g. by letting the entrainment probability increase upon arrival of a moving particle) would change the model behavior.*

We realize that the conditions leading to deposition in the model needed to be explained better. Both mechanisms described by the referee in his comment lead to deposition: a particle can deposit because it arrives in a pocket or because of a collision with another particle (or with the channel banks); when two particles collide they both deposit. This second mechanism is much less important than the first one because particle-particle collisions can only happen when at least one of the particles moves laterally, i.e. when they are trying to occupy the same space. The revised manuscript has been modified and the conditions leading to deposition are now clearly listed (Section 2.1.4 P6 L17-26).

If we have understood correctly the second part of the comment, the referee is asking to take into account what in the literature can be called as "collective entrainment" (e.g. in C. Ancey's work), i.e. the fact than when a particle deposits on the bed it hits the surrounding particles enhancing their chance of being entrained. We agree with the Referee that it would be interesting to test this aspect with a RC approach. However, in the context CAST, this will introduce many additional arbitrary parameters, i.e. how much the entrainment parameter should be decreased? For how many cells in the vicinity of the deposited particle? For how long in time will this effect last? Therefore, we have decided not to address this point in the context of our present study, leaving it for the next development stage of the model. In the revised manuscript we added one paragraph in the outlook addressing this topic (P16 L15-17) and one sentence in section 2.1.2 (P5 L9-11) to clarify that this aspect is not directly accounted for in the current version of our model.

[revised manuscript text omitted]